# Position: The Time for Sampling Is Now!
# Charting a New Course for Bayesian Deep Learning

**Emanuel Sommer** [1] [2]   **David Rügamer** [1] [2]

## Abstract

The practical adoption of sampling-based inference (SAI) in Bayesian neural networks (BNNs) remains limited, partly due to persistent misconceptions about the feasibility and efficiency of sampling. This position paper argues that SAI has achieved computational parity with optimization-based methods and is at the verge of superseding such methods for effective and efficient inference in BNNs. This development should be in the interest of the whole community, promoting BNNs as a principled paradigm with its long-standing yet unfulfilled promise of providing principled uncertainty quantification for neural networks. SAI can even do more—yielding superior prediction performance through model averaging, serving as the foundation for a plethora of possible downstream tasks, and providing crucial insights into the landscape of BNNs. In order to make such a change happen and unfold the potential of sampling, overcoming current misconceptions is a necessary first step. The next step is to realign research efforts toward addressing remaining challenges in SAI. In particular, the community must focus on two core problems: sufficient exploration of the posterior landscape and high-fidelity distillation of posterior samples for efficient downstream inference. By addressing conceptual and practical obstacles, we can unlock the full potential of SAI and establish it as a central tool in Bayesian deep learning.

## 1. Introduction and Motivation

With the widespread use of deep learning-based systems both in science and industry (e.g., Eraslan et al., 2019; Abramson et al., 2024), the need for capturing the epistemic uncertainty of these systems is also rising (Hüllermeier & Waegeman, 2021; Murphy, 2023). Bayesian deep learning (BDL) offers a principled framework to capture this uncertainty through the model's posterior and has led to the development of various approximation methods for neural networks (Papamarkou et al., 2024).

Contemporary research in BDL predominantly follows two directions. One branch focuses on sampling-based inference (SAI), emphasizing asymptotic guarantees while avoiding reliance on simplifying distributional assumptions, but being limited to small-scale problems (Cobb & Jalaian, 2021; Wiese et al., 2023). The other recasts the search for the posterior as an optimization problem, prioritizing scalability and speed at the expense of a more rigid approximation (Blei et al., 2017; Daxberger et al., 2021a; Shen et al., 2024).

Recent advancements in SAI, however, demonstrate that sampling methods have the potential to be computationally tractable also for larger-scale problems and empirically often surpass approximate methods in both performance and uncertainty estimation (Deng et al., 2023; Paulin et al., 2025; Sommer et al., 2025). Complementing these methodological advances, software frameworks such as `blackjax` (Cabezas et al., 2024) and `posteriors` (Duffield et al., 2025) now offer faster and more accessible implementations of core sampling algorithms.

Given these developments, we take the following position: **Sampling-based inference is the new future of Bayesian deep learning, but we need to rethink our research focus and inference workflows**. Moving beyond the pursuit of increasingly narrow algorithmic improvements, we argue that the field's progress now hinges on coordinated efforts to build robust and accessible end-to-end workflows. This requires:

1. Addressing persistent misconceptions that currently hinder the broader acceptance and use of sampling-based methods.

2. Developing more effective strategies for exploring the complex posterior landscape of neural networks, leveraging parallelization and optimization insights.

3. Prioritizing the underexplored yet mission-critical area

[1]Department of Statistics, LMU Munich, Munich, Germany [2]Munich Center for Machine Learning, Munich, Germany. Correspondence to: David Rügamer <david@stat.uni-muenchen.de>.

*Proceedings of the 43rd International Conference on Machine Learning*, Seoul, South Korea. PMLR 306, 2026. Copyright 2026 by the author(s).

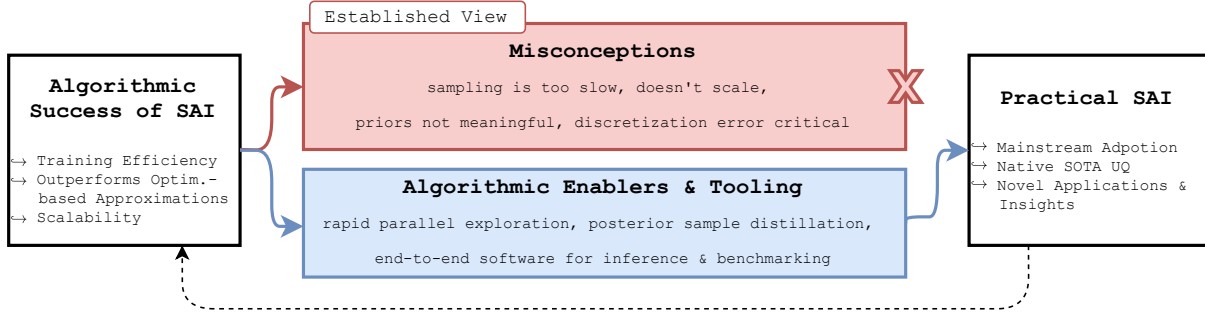

*Figure 1.* Conceptual overview of the pathway toward practical sampling-based inference (SAI). A strong algorithmic status quo already addresses a substantial portion of common misconceptions, while remaining barriers are navigated through additional algorithmic enablers and tooling. Together, these elements enable practical feasibility and widespread adoption. The (dashed) feedback arrow indicates positive momentum from practical SAI, reinforcing further methodological and system-level advances.

of managing posterior samples, including effective storage, distillation, and reuse across various tasks.

Figure 1 offers a visual overview of the status quo, the algorithmic enablers, tooling, and the transformative potential our proposed shift in SAI can unlock. In the remainder of this paper, we articulate this perspective by (1) dispelling common misconceptions about SAI (2) outlining the key design principles and open challenges for scalable posterior sample generation, and (3) how the focus of research should also shift toward developing new methods for practical inference.

## 2. Background

### 2.1. Bayesian Neural Networks

In the Bayesian paradigm, neural network parameters (weights and biases, collected in a flattened vector $\boldsymbol{\theta} \in \Theta \subseteq \mathbb{R}^d$) are treated as random variables and are assigned an explicit prior distribution $p(\boldsymbol{\theta})$. Observing data $\mathcal{D} = \{(\boldsymbol{x}_i, \boldsymbol{y}_i)\}_{i=1}^n \in (\mathcal{X} \times \mathcal{Y})^n$ allows us to update the prior via Bayes' rule, yielding the posterior density

$$p(\boldsymbol{\theta} \mid \mathcal{D}) = \frac{p(\mathcal{D} \mid \boldsymbol{\theta})p(\boldsymbol{\theta})}{p(\mathcal{D})}.$$

If we had access to the posterior, we would be able to quantify the epistemic uncertainty associated with the parameters of the neural network. The posterior predictive density (PPD) for a new observation and label $(\boldsymbol{x}^*, \boldsymbol{y}^*) \in \mathcal{X} \times \mathcal{Y}$ is given by

$$p(\boldsymbol{y}^* \mid \boldsymbol{x}^*, \mathcal{D}) = \int_\Theta p(\boldsymbol{y}^* \mid \boldsymbol{x}^*, \boldsymbol{\theta})p(\boldsymbol{\theta} \mid \mathcal{D})\mathrm{d}\boldsymbol{\theta}$$

and allows quantifying the predictive uncertainty about $\boldsymbol{y}^*$. The key challenge is that the term $p(\mathcal{D})$ in large BNNs is intractable due to $\boldsymbol{\theta}$'s high dimensionality. Note that while

viewing a pretrained model as a prior in transfer or continual learning settings is related and allows casting knowledge transfer as a Bayesian problem, we here focus on the distinct algorithmic challenges of Bayesian neural network training.

**Approximate Bayesian Inference** To circumvent the intractability of the exact posterior, approximate Bayesian inference (ABI) methods typically turn the problem into a much simpler optimization problem, with the approximation typically centered around (a local) Maximum A-Posterior (MAP) estimator of this posterior (see, e.g., Blei et al., 2017). Since the posterior is typically not available in analytic form, the search for a (local) maximizer requires simplifying assumptions. Common ways to make the problem tractable are variational assumptions, where instead of maximizing the actual posterior, a surrogate posterior with simpler structure (e.g., a factorized Gaussian) is used (Ranganath et al., 2014), or an optimized neural network is assumed to be the MAP estimator and a local approximation around this point is made. Common methods include Laplace approximation (Daxberger et al., 2021a), subspace inference (Izmailov et al., 2020; Dold et al., 2025), and stochastic weight averaging (Izmailov et al., 2018). In practice, ensembles of independently initialized and optimized networks called deep ensembles (DE, Lakshminarayanan et al., 2017) constitute a strong and robust baseline for approximating predictive uncertainty, while only being a valid approximation to the posterior in special cases (Wild et al., 2024; Rügamer, 2026). Their Bayesian extension is discussed in this work.

### 2.2. Sampling-Based Inference

Having access to the differentiable prior and likelihood, we can also make use of the unnormalized posterior $p(\mathcal{D} \mid \boldsymbol{\theta})p(\boldsymbol{\theta})$ to construct Markov chains whose stationary distribution is the desired posterior density, building on the rich literature of Markov chain Monte Carlo (MCMC) methods (Gelman et al., 2013). Thus, the approximate posterior in

SAI is characterized by a finite set of $S$ obtained posterior samples $\{\boldsymbol{\theta}^{(s)}, s \in \{1, \ldots, S\}\}$, i.e., a set of neural network weights in the BNN case. When representative and large enough, this set allows for a more flexible and faithful posterior approximation and for estimating properties of the posterior via Monte Carlo estimates. In the case of the PPD, we get

$$p(\boldsymbol{y}^* \mid \boldsymbol{x}^*, \mathcal{D}) \approx \frac{1}{S} \sum_{s=1}^{S} p(\boldsymbol{y}^* \mid \boldsymbol{x}^*, \boldsymbol{\theta}^{(s)}),$$

which constitutes a simple form of a Bayesian model average (BMA) by combining $S$ model parametrizations.

Historically, the complexity of BNN posteriors posed significant hurdles for SAI. Common sampling algorithms often struggled to produce meaningful samples, frequently getting stuck in low-probability regions. More sophisticated samplers, such as Hamiltonian Monte Carlo (HMC, Neal, 2011; Duane et al., 1987) or the No-U-Turn Sampler (NUTS; Hoffman & Gelman, 2014), showed some success but suffered from very slow, ineffective performance in high dimensions and difficult algorithm configuration. Additionally, the requirement for full-batch training in many samplers further impeded their use with large datasets.

## 3. Persisting Misconceptions and Recent Advancements

Recent advances in both core sampling methodology and efficient implementations have substantially alleviated many of the previously existing hurdles. This progress positions SAI comparably to optimization-based approximate inference methods in terms of time complexity and scalability, while demonstrably yielding superior performance (see, e.g. Figure 2). Despite these significant advancements, persistent misconceptions about SAI extend beyond the stereotypical focus on computational inefficiency. Many of these lingering beliefs stem from a naive transfer of techniques from classical MCMC to the distinct context of BNNs. The following discussion addresses the most relevant misconceptions prevalent in the community, specifically those concerning time complexity, scalability, prior choices, (un)adjusted sampling, and effects like the "cold posterior effect".

**Time Complexity** A common perception is that SAI is very slow, making it practically infeasible. Recent comparisons between Bayesian and non-Bayesian DL approaches (see, e.g., Sommer et al., 2025; 2026a; Arvanitis et al., 2026; Paulin et al., 2025; Deng et al., 2023), however, demonstrate that once a good starting value has been determined for each chain, the time for sampling is roughly the same as the initial optimization step. In contrast to runtimes with less advanced methods and software, where sampling costs were 10 or 100 times larger, this clearly indicates a turning

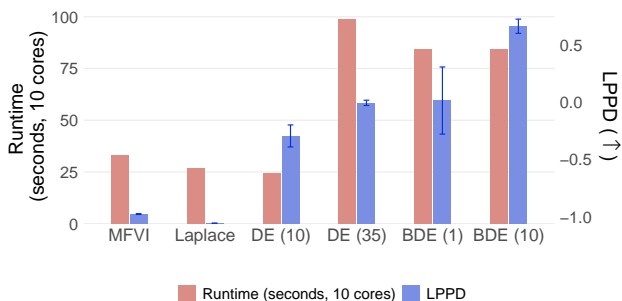

*Figure 2.* Exemplary comparison of runtime (in seconds, using 10 CPU cores) and log pointwise predictive density (LPPD; higher is better) across different Bayesian inference techniques for the `airfoil` UCI regression task. Methods include mean-field variational inference (MFVI), Laplace approximation, and deep ensembles (DE) with 10 and 35 members. SAI is represented by the Bayesian deep ensembles (BDE Sommer et al., 2024) with 1 and 10 Markov chains and 1000 samples each. Error bars indicate standard errors across 3 seeds. The architecture is a three hidden layer MLP with 16 neurons each. BDE consistently achieves higher LPPD than competing methods, even when they are granted the same or greater computational budget. To match the performance of one chain, it requires 35 DE members, which, however, have longer runtimes despite parallelization.

point in sampling-based research. Given that sampling has comparable costs to optimization, most practitioners are likely willing and able to spend this additional time. This, particularly, holds if sampling provides considerably better performance even when competing methods are granted the same or more computational budget, as shown in Figure 2.

**Scalability** Another perception in the community is that sampling techniques cannot scale to high-dimensional problems. However, numerous samplers have proven effective in high-dimensional BNNs (Welling & Teh, 2011; Chen et al., 2014; Springenberg et al., 2016; Zhang et al., 2020; Paulin et al., 2025; Sommer et al., 2026a). Although current use cases of SAI typically involve only millions of parameters, we contend that scaling to even larger models is feasible. The key not only lies in algorithmic advancements but in their synergistic combination with effective exploration techniques (Section 4) and strategies for managing the generated samples (Section 5).

**Prior Choices** Another case in point is the standard isotropic Gaussian prior. While often viewed as a pointless and too simple default, recent work suggests it may have desirable regularizing properties in the overparameterized regime, possibly contributing to implicit bias effects similar to those exploited in frequentist deep learning (Fortuin, 2022; Tran et al., 2022; Kobialka et al., 2026). Moreover, contrary to these recurring concerns, various recent papers have shown state-of-the-art performance of BNNs when using these simple weight priors (Kobialka et al., 2026;

Paulin et al., 2025; Kim et al., 2024). Given that independent weight initializations are also a well-working concept in optimization-based approaches, we argue that priors are a reasonable and well-suited choice for BDL.

**Metropolis-Hastings Adjustment**   While the discretization error in stochastic gradient-based samplers has received considerable attention (Cobb & Jalaian, 2021), it often represents a subdominant source of approximation error. This highlights a bias-variance trade-off that is specific to the practical BNN setting. While strictly eliminating discretization error (often via Metropolis-Hastings adjustment) is standard practice in general MCMC to ensure detailed balance, doing so in high dimensions incurs disproportionate inefficiencies. It often leads to extremely low acceptance rates and, consequently, large Monte Carlo error under any realistic computational budget. Therefore, in practical regimes where a sufficiently low step size is used, initialization error and Monte Carlo error from insufficient samples typically dominate over discretization bias (Neal, 1992; Roberts & Tweedie, 1996; Welling & Teh, 2011; Robnik & Seljak, 2024b; Robnik et al., 2025; Sommer et al., 2025). Moreover, in high-dimensional settings such as for BNN inference, the sampling path is confined to a localized one, further reducing the importance of fine-grained discretization control. While controlling discretization fidelity is necessary to ensure efficient sampling, strict adjustment for discretization bias will not be key to unlock the full potential of SAI in high dimensions, as also suggested in Bieringer et al. (2023); Garriga-Alonso & Fortuin (2021) and for some samplers, specific tunings can be employed to keep the discretization error low in a scalable, lightweight manner (Sommer et al., 2026a).

**Interaction of Key Design Choices**   Many misconceptions exist because interactions between the core design choices of sampling approaches like *prior specification*, *sampler noise injection*, and the resulting *posterior contraction* have been overlooked. For example, the classical bias-variance trade-off—where excessive noise results in biased estimates and insufficient noise leads to poor exploration—has been discussed in both theory and practice (Wenzel et al., 2020; Robnik & Seljak, 2024a). Yet in the context of BNNs, key components such as priors, data augmentations, and sampling schemes are often studied in isolation. For instance, the perceived "cold posterior effect", where posterior tempering improves performance, was initially hypothesized to be connected to a misalignment between likelihood and prior. However, it was later shown that much of this effect can be attributed to underappreciated interactions with data augmentation and the choice of the likelihood in classification settings, rather than a flawed target distribution (Wenzel et al., 2020; Izmailov et al., 2021; Bachmann et al., 2022; Kapoor et al., 2022).

By untangling these design choices and moving past outdated misconceptions, the community can focus on the core benefits of SAI, which include faithful uncertainty quantification, a principled foundation for downstream tasks, and deeper structural insights into BNN posteriors.

## 4. Scalable Sampling for Bayesian Neural Networks

Building on the previously introduced foundations and recent developments in the field, the following outlines our position on the most important future directions to achieve robust, effective, and scalable sampling in Bayesian neural networks.

### 4.1. Flexible Exploration Over Restrictive Approximation

> BDL research should focus on an **efficient and flexible exploration** of the posterior space, **rather than** on refining **posterior approximations** that cast Bayesian inference as an optimization problem using rigid assumptions.

The BDL community appears increasingly polarized: on one side, researchers pursue sophisticated but computationally demanding methods that only scale to small problems; on the other, practitioners working with large-scale problems often rely on too restrictive approximations such as fully factorized variational families. Recent developments introduce specific inductive biases that address narrow subproblems, yet often lack general applicability or scalability. Techniques such as adversarial optimization have been proposed to steer inference toward flatter minima (Lim et al., 2025), where approximations may be more faithful. Recent empirical studies, however, suggest that such approximations still fall short in predictive performance and uncertainty quantification compared to SAI (Kobialka et al., 2026).

Despite its limitations, approximate Bayesian inference remains a dominant paradigm largely due to its familiarity and simplicity in implementation. However, the more complex the refinements, the less likely such methods are to be adopted in practice, especially when they introduce additional hyperparameters or necessitate careful tuning. In practice such methods are therefore often employed only in their most rudimentary form (see, e.g., Kaiser et al., 2025).

This highlights the imperative to redirect BDL research priorities: rather than optimizing within a restrictive space, we advocate for the development of scalable SAI methods that retain full posterior flexibility. SAI avoids the structural limitations of variational families and offers the potential for truly flexible epistemic UQ.

Moreover, we argue for a more strategic evaluation of ap-

*Table 1.* Meta-study of SAI scalability and performance. We synthesize exemplary empirical results from recent literature to demonstrate that SAI consistently achieves parity with or superiority over strong optimization baselines (DE, SWA, SGD) across a vast range of model scales. While the table highlights comparisons against ensemble and optimization baselines, the underlying study for the ResNet-18 experiments (Sommer et al., 2026a) additionally reports that SAI outperforms both variational inference and Laplace approximation on these metrics.

| Experiment | Architecture | Params | Task | Metric | Baseline | SAI |
|---|---|---|---|---|---|---|
| **Standard Vision** Sommer et al. (2026a) | ResNet-18 | 11.2M | CIFAR-10 | LPPD ↑ Accuracy ↑ | $-0.30$ (DE) 0.90 (DE) | $-$**0.26** (pSMILE) **0.91** (pSMILE) |
| **Mid-Scale ViT** Sommer et al. (2026a) | Vision Transformer | 22M | Imagenette | LPPD ↑ Accuracy ↑ | $-0.99$ (DE) **0.77** (DE) | $-$**0.77** (pSMILE) **0.77** (pSMILE) |
| **Large-Scale Vision** Paulin et al. (2025) | Wide CNN | 73M | CIFAR-10 | NLL ↓ Calib. (ACE) ↓ | 0.56 (SWA) 0.014 (SWA) | **0.31** (SMS-UBU) **0.005** (SMS-UBU) |
| **LLM finetuning** Duffield et al. (2025) | Llama 3 | 218M (8B total) | OOD Detect | AUROC ↑ Loss (English) ↓ | 0.80 (SGD) **4.20** (SGD) | **0.95** (SGHMC) 4.40 (SGHMC) |

proximate inference methods on problems that are both practically relevant and computationally manageable. Instead of benchmarking on toy models with tens of parameters or resorting to billion-scale models where only crude approximations are feasible, the community should explore the middle ground: modern, mid-scale architectures where scalable yet flexible inference is both necessary and feasible. Concrete examples of this regime include ResNets, Vision Transformers, and large language models (LLMs) fine-tuning adapters, with parameter counts ranging from hundreds of thousands to tens of millions (see, e.g., Paulin et al., 2025; Duffield et al., 2025; Sommer et al., 2026a). As summarized in Table 1, this emerging evidence provides a compelling proof-of-concept, demonstrating that SAI is not only feasible at this scale but can consistently outperform alternative, purely optimization-based approximations and strong baselines like stochastic weight averaging (SWA), Laplace approximation, variational inference, and DE.

### 4.2. Parallelized Exploration is Key

> **Efficient high-dimensional sampling should parallelize the exploration phase and leverage optimization techniques** to rapidly approach the typical set, instead of relying on inefficient sequential exploration schemes.

The posterior distribution of neural networks is known to be complex and highly multimodal (Sommer et al., 2024; Izmailov et al., 2021). To improve the exploration capabilities of samplers, various strategies have been proposed—ranging from tempered Langevin dynamics (Welling & Teh, 2011; Li et al., 2023) to cyclical step sizes in stochastic gradient Langevin dynamics (Zhang et al., 2020). These methods typically aim to enhance exploration in a sequential setting by alternating between phases of coarse and fine approximation. In high-dimensional settings, however, sequential exploration becomes increasingly inefficient due to increased autocorrelations (Papamarkou et al., 2022). While earlier

foundational studies were computationally limited to exploring a small number of sequential chains (Izmailov et al., 2021), recent work on the exact same model classes places considerably higher emphasis on the necessity and merits of parallel exploration (Marek et al., 2024). In particular, recent work demonstrates that parallel, optimization-driven exploration offers superior computational performance and sampling quality compared to purely sequential approaches (Deng et al., 2023; Rundel et al., 2025; Duffield et al., 2025; Sommer et al., 2025; Paulin et al., 2025). DEs have already highlighted the benefits of parallel and independent exploration (Lakshminarayanan et al., 2017; D'Angelo & Fortuin, 2021) and have long been considered a state-of-the-art method for epistemic uncertainty quantification.

While DEs do not represent a valid posterior approximation (Wild et al., 2024), they can provide a robust and computationally efficient warm-start for sampling-based inference (Sommer et al., 2024; 2025; 2026a). Furthermore, optimization-based methods and concepts are increasingly integrated into sampling routines, yielding improved efficiency in navigating complex posterior landscapes (Springenberg et al., 2016; Bieringer et al., 2023; Leimkuhler et al., 2025). For example, Paulin et al. (2025) demonstrate that increasing the number of parallel chains reduces sensitivity to local approximation errors, confirming that the main bottleneck in sampling-based inference is not necessarily the design of better samplers, but rather the lack of scalable exploration strategies.

We argue that parallelized, optimization-informed exploration is essential for effective sampling-based inference. In particular, a rapid initial exploration phase—guided by robust optimization techniques—allows for starting chains within meaningful proximity of the typical set, thus enabling efficient and reliable sampling.

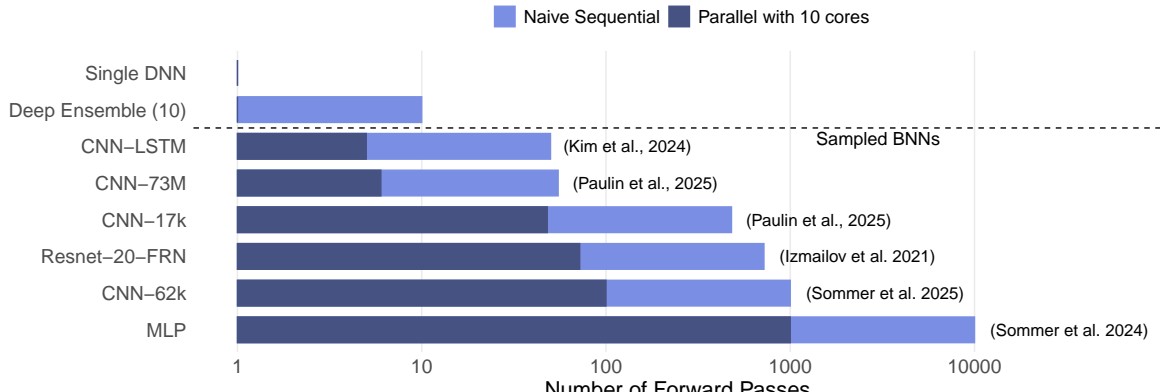

*Figure 3.* Illustrative comparison of inference costs for state-of-the-art BNNs in terms of the number of forward passes required for a single batch of test points. Models and sample counts are taken from recent works (Izmailov et al., 2021; Kim et al., 2024; Paulin et al., 2025; Sommer et al., 2024; 2025) and span a range of architectures from small MLPs to large-scale CNNs with tens of millions of parameters. As a baseline reference, we report the inference costs for a single deep neural network (DNN) and an ensemble of 10 DNNs. To indicate practical wall-clock time, the 'parallel' bars (dark blue) assume distribution across 10 CPU cores (dividing total passes by 10).

## 4.3. The Untapped Potential of SG-MCMC

> The **potential of SG-MCMC remains largely untapped**, hindered by limited empirical guidance on practical algorithm configuration.

Despite the growing availability of efficient and scalable stochastic gradient SG-MCMC implementations such as `posteriors` (Duffield et al., 2025), Fortuna (Detommaso et al., 2023), `JaxSGMC` (Thaler et al., 2024) and `SGMCMCJax` (Coullon & Nemeth, 2022), these methods remain difficult to deploy. In particular, all SG-MCMC methods are highly sensitive to the many different hyperparameters, such as batch and step sizes or noise levels. Even adaptive strategies, such as those proposed in Springenberg et al. (2016) for SG-HMC, offer only partial relief.

To fully realize the promise of SG-MCMC in all its variations, from overdamped dynamics (Ma et al., 2015) over parallel tempering (Deng et al., 2023) to samplers without damping (Sommer et al., 2026a), the field must invest in large-scale empirical studies and foster open sharing of tuning insights as well as failure modes of SG-MCMC. Rather than treating configuration as a black art, the community should collaboratively develop transparent heuristics and best practices tailored to specific problem settings.

## 5. Efficient and Accessible Inference

We are absolutely convinced that sampling can be scaled to large-scale DL problems. However, this is only half the battle, as illustrated by the inference cost comparison of state-of-the-art BNNs in Figure 3. If research does not start exploring ways to efficiently store, distill, and reuse posterior samples, even the most sophisticated sampling procedure will not be adopted by practitioners if the obtained information cannot be used efficiently and effectively.

## 5.1. Store First, Improve Later

> Saving samples requires an upfront cost to retain all information, but **is essential and only requires cheap hard disk memory**. This should be the preferred option over naive thinning techniques.

A common but suboptimal strategy for managing the storage and computational overhead of sampling is thinning (Duffield et al., 2025; Kim et al., 2024; Izmailov et al., 2021), a point increasingly acknowledged in classical Bayesian MCMC (Riabiz et al., 2022). We argue that, except for extremely large BNNs where online methods are necessary, the default should be to store most or all generated samples. Efficient post-sampling distillation techniques can then be applied for storage reduction and/or faster inference, preserving valuable information that is lost through indiscriminate thinning. We think that the optimal use of posterior samples varies depending on the downstream task and the evaluation metric of interest (e.g., mean prediction requires fewer and possibly different samples than accurate tail probability estimation). Therefore, the SAI community should avoid both naive thinning and an exclusive focus on single evaluation metrics.

Furthermore, retaining a comprehensive set of posterior samples provides invaluable data for analyzing sampler dynamics and assessing (local) convergence. As a tailored evaluation framework for BNNs is critically needed (Rønning et al., 2025), the availability of more samples facilitates a deeper understanding and more precise quantification of these aspects, directly aiding sampler development.

## 5.2. More Efficient Storage and Inference

> **Distilling posterior samples will be a key strategy to drastically improve inference efficiency** while enhancing downstream performance and calibration. To support this, the community should build benchmark collections of posterior samples to foster reproducibility and empirical progress.

Inference-time inefficiency is not merely a computational nuisance. It is the key practical bottleneck once efficient sampling routines have been established. For each batch of new data, we perform a forward pass for each generated posterior sample, which also needs to be available in memory. Although this can be embarrassingly parallelized, we show in Figure 3 that even with parallelization, the inference cost for modern sampled BNNs can be up to 1000 times as expensive as vanilla neural networks or fully linearized approaches (Li et al., 2025). This does not even consider the memory access and I/O overhead if all samples cannot be kept in memory simultaneously.

Relying on unprincipled and uninformed methods such as thinning discards samples indiscriminately and fails to address the root cause of redundancy in posterior samples. Therefore, the next important research direction in SAI is how to distill posterior samples as a post-hoc optimization stage that compresses a large set of posterior samples into a smaller, more informative set. This process can simultaneously reduce the number of forward passes required at test time and improve the quality of predictions by implicitly reweighting or summarizing important regions of the posterior. Such methods can thereby also counter issues of autocorrelation and mode imbalance.

Approaches for posterior samples distillation are method-agnostic and can be instantiated through a wide spectrum of techniques, including classic data compression, sequential testing (see, e.g., Sommer et al., 2026b), learned generative surrogates in weight space (see, e.g., Park et al., 2025), or principled sub- or importance sampling strategies. These methods complement the base sampler and directly target the dominant cost driver in practical BNN inference: processing large numbers of network instantiations at test time. Moreover, such techniques can be specifically targeted towards a wide and diverse range of goals and metrics, depending on the downstream usage such as calibration.

Beyond post-hoc posterior distillation, techniques such as Bayesian coresets (Huggins et al., 2016) or dataset distillation (Wang et al., 2018) can substantially reduce the computational burden per gradient step. This is particularly beneficial for extending the feasible regime of classic full-batch MCMC and bridging the gap to the SG-MCMC methods discussed above.

We further suggest that the community develop a standardized way of collecting posterior samples to serve as benchmarks for testing the performance of posterior sampling and distillation methods.

## 5.3. Smarter Bayesian Model Averaging

> **Smart aggregation and averaging** schemes not only improve efficiency but can potentially also **lead to significant performance boosts**.

Recent findings show that even small Bayesian deep ensembles that leverage ensembled posterior sampling can surpass the performance of much larger DEs (Kobialka et al., 2026) as also depicted in Figure 2. The advantage stems from a meaningful and flexible local exploration, rather than the sheer size of the ensemble.

The effectiveness of DEs arises not merely from ensembling but from their ability to strike a balance between exploration and exploitation during training. DEs also yield a diverse set of solutions that, when averaged, form a powerful approximation to a BMA. However, standard MAP ensembling saturates quickly and lacks principled weighting and a sense of local uncertainty (Wild et al., 2024; Wang & Wang, 2025). Approaches such as subspace inference (Izmailov et al., 2020; Dold et al., 2024; 2025) or SWA-based initialization (Izmailov et al., 2018; Paulin et al., 2025) further support the view that better exploitation of diversity guided by posterior structure leads to stronger generalization and more reliable uncertainty.

These findings demonstrate that the power of BMA in SAI is far from exhausted, and even straightforward extensions like sample weighting offer avenues to further enhance predictive quality and calibration (Yao et al., 2022; Sheinkman & Wade, 2025).

## 5.4. Software

> **The post-sampling phase**, i.e., posterior distillation, serving, and sharing, **demands dedicated tooling** to unlock the full potential of sampling-based Bayesian inference for neural networks.

Despite recent advances in posterior sampling, the surrounding ecosystem lacks mature end-to-end tooling tailored to the demands of scalable Bayesian neural networks. In contrast to established frameworks for classical Bayesian modeling (PyMC; PyMC-Devs, 2025) or general-purpose deep learning, e.g., Keras (Chollet et al., 2015), practitioners have limited access to performant and modular tools for managing large posterior collections and processing them for downstream deployment.

We advocate for building performant libraries that integrate modular design, automatic optimization, and principled default behaviors. These tools should prioritize model serving efficiency and be equipped to handle the posterior distillation pipeline. Recent developments in core sampling libraries (cf. Section 1) provide a foundation, but broader infrastructure is needed. For example, a highly effective pipeline should utilize asynchronous callbacks that eagerly save samples to disk. This approach incurs almost no relevant I/O overhead and prevents memory accumulation, allowing hardware accelerators to be fully utilized without resorting to overly aggressive thinning intervals at the cost of information loss.

Furthermore, diagnostic software like the `ArviZ` library (Kumar et al., 2019), which is limited to classical Bayesian models and diagnostics, should be extended for the BNN use case. The current reliance on classical convergence diagnostics in BNN research, whether in parameter (e.g., Jawla & Kelleher, 2025) or function space (e.g., Andrade & Sato, 2024; Fortuin, 2022), is often problematic: parameter space diagnostics are uninformative for non-identifiable BNNs, and function space metrics can promote misleading homogeneity (Sommer et al., 2024). Developing better, tailored evaluation frameworks for BNN convergence is still an open and highly relevant issue.

Standardized interfaces like `huggingface` for posterior sample sharing would further promote reproducibility, empirical benchmarking, and community-driven progress. Making posterior samples accessible, comparable, and efficiently deployable is a necessary step toward mainstream adoption of sampling-based Bayesian deep learning.

## 6. Alternative Views

Before concluding our position, we briefly list some of the alternative views and common criticisms that are raised in the discussion about the practicability and efficacy of SAI.

### 6.1. Classical Approximate Bayesian Perspectives

**Alternative view: Approximate Bayesian methods are closer to the standard workflow in DL and hence more practical than SAI.** Our response: We agree with this point of view. This is precisely what we advocate for. The perceived gap exists due to a current lack of robust end-to-end software for SAI (Section 5.4) and a significant discrepancy in how inference is managed (Section 5.2). We argue that by building and sharing empirical knowledge, similar to how classical DL matured, we can make SAI workflows equally accessible and intuitive (Section 4.3).

**Common criticism: Practitioners can never be sure whether sampling reaches a stationary distribution.** Our response: No one can guarantee convergence in practice under realistic computational budget constraints in high-dimensional BNNs. Thus, SAI also clearly remains an approximate method. However, the flexibility of sampling-based approximations demonstrably outperforms the more rigid structural assumptions of other approximate Bayesian methods, yielding superior predictive and uncertainty quantification performance in many cases (Section 4.1). Our goal is not asymptotic perfection, but a substantially better and more trustworthy approximation.

**Alternative view: Approximate Bayesian methods will always be at least as scalable as SAI.** Our response: While approximate Bayesian methods might have an inherent scalability advantage in certain aspects, if the computational cost for SAI is comparable, the significantly improved predictive performance and uncertainty quantification offer a compelling trade-off. We posit that the future lies in leveraging optimization and approximate Bayesian tools to *boost* the scalability of SAI, particularly through parallelized exploration strategies (Section 4.2), rather than viewing them as fundamentally separate or competing paradigms.

**Alternative view: Some approximate Bayesian methods can be proven to recover important characteristics.** Our response: Such theoretical guarantees (see, e.g., Margossian & Saul, 2025) often rely on strong, sometimes unrealistic, assumptions that frequently break down in the complex, multimodal posterior landscapes typical of BNNs. Our focus on flexible sampling approaches aims to capture the full posterior structure, which is often beyond the reach of "guaranteed" yet overly restrictive approximations.

### 6.2. Foundation Models

**Common criticism: No one will ever be able to store multiple weight copies of a GPT-5 model.** Our response: For models on the scale of GPT-5, the organizations capable of training such models will also possess the resources to store multiple weight copies or posterior samples. For the broader community, training and storing models of this magnitude is already infeasible even in a non-Bayesian context. Our position primarily targets a vast range of modern, mid-to-large scale BNNs (as, e.g., in Table 1) where storing samples is practical and beneficial, as discussed in Section 5.

**Skeptical question: Is the computational overhead of Bayesian Deep Learning justified for LLMs?** Our response: As LLMs cement their role as the backbone of many modern AI systems; it is natural to question the relevance of quantifying epistemic uncertainty through BDL in this regime. To answer whether the computational overhead

of a Bayesian treatment is justified for foundation models, we must distinguish between *semantic* and *epistemic* uncertainty. In open-ended generation, ambiguity is inherent to the data; the model may be certain about the distribution of plausible next tokens, yet the semantic outcome varies naturally (Kuhn et al., 2023). While experiments suggest that Bayesian marginalization can improve likelihood metrics such as perplexity (Sommer et al., 2026a), the practical utility of quantifying and incorporating epistemic uncertainty in highly stochastic, open-ended generation tasks remains an active area of research, and its exact benefits are still unclear.

However, a significant portion of industrial LLM deployment—ranging from automated triage and medical diagnosis support to RAG context optimization—relies on LLMs acting as discriminative classifiers (Brown et al., 2020). In these supervised settings, the model acts as a decision maker rather than a creative writer. Here, the ability to quantify epistemic uncertainty via the posterior predictive density is well-posed. We envision token-level uncertainty as particularly impactful in the context of structured generation (Willard & Louf, 2023)—a paradigm essential for agentic workflows where reliable, syntactically correct outputs are prerequisites for effective autonomous action.

While sampling billions of parameters is currently infeasible for most practitioners, this constraint applies equally to standard optimization; full pre-training or fine-tuning on this scale is rarely performed. We argue that SAI need not imply full-model sampling. Recent successes in sampling low-rank adapters (Duffield et al., 2025; Yang et al., 2024) or restricting inference in classical BDL to specific subnetworks (Daxberger et al., 2021b) demonstrate that we can treat pre-trained weights as a fixed feature extractor and perform efficient sampling on a specific subset. This hybrid approach offers a pragmatic path to robust UQ in foundation models without the potentially prohibitive costs of full-scale MCMC.

## 7. Conclusion

Sampling-based inference for Bayesian neural networks has reached a critical juncture. While advancements in algorithms demonstrate its efficiency and competitiveness, the field must now pivot its attention towards the practicalities of deployment. Our central position is that the key to widespread adoption lies not solely in algorithmic improvements but in tackling the under-addressed challenges of an end-to-end sampling pipeline, while also dispelling some enduring misconceptions about the initial sampling cost. This includes embracing methods to efficiently generate and manage posterior samples (countering the notion that this phase is inherently prohibitive) and developing effective strategies to distill posterior samples for rapid inference. By aban-

doning outdated assumptions rooted in classical Bayesian settings and collaboratively investing in both clarifying the feasibility of sample generation and the underexplored area of downstream sample utilization, the BNN community can transform sampling from a theoretical ideal into a practical and impactful reality for Bayesian deep learning.

## Acknowledgements

The authors are grateful to the anonymous reviewers for their thorough evaluation and valuable suggestions, which helped refine the arguments presented in this work.

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

## A. Experimental Details

**Runtime Illustration**  To calculate the runtime results shown in Figure 2 we were using the code from Kobialka et al. (2026). We adopt the same UCI benchmark setting as in their Table 2, use the `airfoil` dataset (Dua & Graff, 2017) and extend their analysis by reporting average runtimes measured on a standard 10-core CPU. To ensure a fair comparison under equal computational budgets, we additionally include a 35-member Deep Ensemble—the most performant competing method.

## B. Clarification on Terminology: SAI vs. SBI

Throughout this paper, we utilize the acronym SAI to denote Sampling-Based Inference. This refers to the broad class of methods that generate samples from a posterior distribution $p(\boldsymbol{\theta} \mid \mathcal{D})$ where the likelihood function $p(\mathcal{D} \mid \boldsymbol{\theta})$ is explicit and differentiable, which is the standard setting for Bayesian Neural Networks trained via backpropagation.

We distinguish this explicitly from SBI (Simulation-Based Inference), also known as Likelihood-Free Inference (Cranmer et al., 2019). SBI addresses scenarios where the likelihood is intractable or implicitly defined by a stochastic simulator (e.g., in physics or epidemiology). While SBI often utilizes neural networks (e.g., via Normalizing Flows in Neural Posterior Estimation) to approximate posteriors, the fundamental challenge there is the absence of gradients from the likelihood.

In contrast, the position presented in this work targets the regime where gradients are available and cheap (standard Deep Learning), but the high dimensionality and multimodality of the parameter space $\Theta$ make exploration difficult. We deliberately use SAI to avoid conflation with the likelihood-free constraints of the SBI literature.

