# OpenReview forum: "Position: The Time for Sampling Is Now! Charting a New Course for Bayesian Deep Learning"
_ICML.cc/2026/Position_Paper_Track — ICML 2026 Position Paper Track spotlight_

### Official Review · Reviewer_mVE6 · 2026-02-27

**Significance:** 2
**Argument Clarity:** 2
**Rating:** 5
**Confidence:** 4

**Questions:**

1. In Section 4.2, the authors argue that parallelized exploration should be prioritized over sequential exploration. This seems to contrast with the empirical conclusions of [Izmailov et al. (2021)](https://proceedings.mlr.press/v139/izmailov21a.html) regarding comparisons between a single HMC run and multiple chains; it might be worth commenting upon this.

2. The discussion of the "untapped potential of SG-MCMC" in Section 4.3 is rather brief. I think it would be useful to include a broader discussion of the design space of MCMC algorithms, including Hamiltonian dynamics and beyond (see for instance [Ma, Chen, and Fox, NeurIPS 2015](https://arxiv.org/abs/1506.04696)). Given the vast design space at the dynamical level, it would be useful for the authors to comment on the potential scalability of different approaches.

3. The authors argue that it should be standard practice to store most or all generated parameter samples, on the basis that it "only requires cheap hard disk memory". To make this practicable, I suspect one would need a specialized caching toolchain that ensures that the sampling algorithm is not rate-limited by time required to write samples to disk. It would be useful to comment (in rough terms) about the expected computational overhead of storing all samples. Also, given the paper's general call for specialized SAI tooling and workflows, it would be useful to concretely comment on what would be required here.

**Alternative Views Section:**

Yes

**Compliance With Llm Reviewing Policy A Conservative:**

Affirmed.

**Discussion Potential:**

3

**Final Justification:**

The authors' replies adequately addressed my questions, and I raised my score from 4 to 5. I was already in favor of acceptance, and their response solidified that assessment.

**Paper Summary:**

This position paper argues that the community of researchers working on Bayesian deep learning should focus their efforts on sampling-based inference, in particular in preference to alternative approximate Bayesian methods (like variational approximations). The central thesis of the work is that, with concerted research effort, it will be possible to overcome many of the current (perceived and actual) limitations of sampling-based inference.

I note that, throughout my review, I will adopt the author's abbreviation of "sampling-based inference" as "SAI". I will abbreviate "Bayesian deep learning" as "BDL".

**Position:**

Yes

**Position In Title:**

Yes

**Related Work:**

3

**Strengths And Weaknesses:**

Debates surrounding the use and future of BDL are clearly of relevance and importance to the ICML community, as evidenced by the publication of a previous position paper (on the broader issue of whether BDL in the large holds promise) by [Papamarkou et al.](https://proceedings.mlr.press/v235/papamarkou24b.html) at ICML 2024. The present paper's strength is its focus on a concrete issue that will help determine this future: the scalability of SAI. I am therefore generally in favor of acceptance, but I think some revision is required.

Where the authors focus on particular questions of SAI, I think their arguments are generally clear and mostly supported by relevant citations. I found the discussion of "Epistemic Uncertainty in LLMs" in Section 6 somewhat weaker and tangential to the main thrust of the argument; I would suggest that the authors re-apportion these points to the Introduction and to related "common criticisms" in Section 7. This would help maintain focus on the main argument regarding SAI, rather than tangential arguments about BDL and uncertainty in the large.

As it is key to their argument, I also wish the authors devoted more space in Section 3 to addressing why they think it is a misconception that SAI has substantially greater time complexity and inferior scalability to high dimensions than other methods. These issues are of course related, which the authors could make clearer. When they discuss time complexity, they rely on a single recent citation, which is not very compelling. It would be helpful to provide a more detailed history of viewpoints on these issues, and stronger evidence for why the authors believe it can be overcome. This, I think, is the main change the authors could make that would strengthen the argument in favor of their position.

Beyond these basic concerns, I list under **Questions** a few points which the authors might consider addressing.

**Support:**

3

---

> ### Author Rebuttal · Authors · 2026-03-30
>
> Dear Reviewer mVE6,
>
> Thank you for your constructive feedback. We appreciate your support for acceptance and your concrete suggestions for strengthening the paper.
>
> **Section 6 Restructuring.**
>
> > I found the discussion of "Epistemic Uncertainty in LLMs" in Section 6 somewhat weaker and tangential. I would suggest that the authors re-apportion these points to the Introduction and to related "common criticisms" in Section 7.
>
> We thank the reviewer for the suggestion and will redistribute the LLM-specific points to the Introduction and to the relevant alternative views in Section 7.
>
> **Time Complexity Evidence.**
>
> > When they discuss time complexity, they rely on a single recent citation, which is not very compelling. It would be helpful to provide a more detailed history of viewpoints.
>
> Thank you for emphasizing this important point. The results in Sommer et al. (2025b) have been independently validated in several other recent works: Paulin et al. (2025, AISTATS), Duffield et al. (2025, ICLR), Liang et al. (2025, Scalable Bayesian Monte Carlo: fast uncertainty estimation beyond deep ensembles, NeurIPS Workshop), and Deng et al. (2023, Non-reversible Parallel Tempering for Deep Posterior Approximation, AAAI). We will add and discuss these in more detail.
>
> **Parallelized vs. Sequential Exploration.**
>
> > This seems to contrast with the empirical conclusions of Izmailov et al. (2021) regarding comparisons between a single HMC run and multiple chains.
>
> Due to the massive computational requirements, Izmailov et al. (2021) could only explore an ensemble of at most 3 parallel chains. The same group placed considerably higher emphasis on parallel exploration in follow-up work (Marek et al., 2024). Together with the recent papers mentioned above, we believe the merits of parallel exploration are now well established. We will extend the discussion accordingly and thank the reviewer for raising this point.
>
> **SG-MCMC Design Space.**
>
> > I think it would be useful to include a broader discussion of the design space of MCMC algorithms, including Hamiltonian dynamics and beyond (see Ma, Chen, and Fox, NeurIPS 2015).
>
> This is an excellent suggestion. Ma et al. (2015) provides a great unifying reference for the popular class of overdamped SGMCMC samplers. We will extend the discussion to also cover parallel tempering approaches and samplers that do not require explicit damping. This broader perspective on the design space will strengthen the argument about the untapped potential of SGMCMC.
>
> **Storage Overhead.**
>
> > I suspect one would need a specialized caching toolchain [...] It would be useful to comment about the expected computational overhead.
>
> This is a very perceptive observation. In our experience, an asynchronous callback that eagerly saves samples to disk (without batching or accumulating in memory) incurs almost no relevant overhead while keeping memory requirements low so that GPUs can be fully utilized. For smaller models, batching writes could make sense, but in these settings the I/O overhead is even more negligible. In practice, one would typically not save after every gradient step but perhaps every 10th or so. Our argument is that very aggressive thinning intervals (e.g., saving only every 50 epochs) discard too much information and should be avoided. We further argue that the most impactful tooling would cover exactly this pipeline: efficient eager sample saving, followed by post-hoc distillation and compression, and finally efficient model serving/sharing with built-in parallelization. In terms of workflows we identify evaluation/diagnostics as the most impactful avenue to contribute specialized SAI software. We will add a concrete discussion of these practical considerations.
>
> ___
>
> We believe the reviewer's suggestions will notably strengthen the paper. Please let us know if there are any further points we can address. We thank the reviewer again for the constructive feedback!

---

> > ### Author Rebuttal · Reviewer_mVE6 · 2026-03-31
> >
> > Thank you for your detailed replies to my comments, and to those of the other referees. I think my concerns have been adequately addressed, and I have accordingly raised my score to "accept".

---

### Official Review · Reviewer_Wk62 · 2026-03-06

**Significance:** 3
**Argument Clarity:** 4
**Rating:** 5
**Confidence:** 4

**Questions:**

I am a bit confused by Figure 3:
- How is parallism employed that it changes the number of forward passes? Or are you simply dividing the number by 10 to mean something like "number of forward passes per unit time?"
- The number of forward passes will depend on the number of posterior samples, right? Which values were chosen in the different papers you reference in the figures. And why? This info may be critical to understand the validity and significance of the figure.

Minor:
- The abbrevition "DE" was introduced, I think, only in a figure caption. Could this be introduced more prominently somewhere else too?
- Why not write end-to-end instead of E2E or is this formulation so common that it merits not writing a few more symbols for the full words?

**Alternative Views Section:**

Yes

**Compliance With Llm Reviewing Policy A Conservative:**

Affirmed.

**Discussion Potential:**

4

**Paper Summary:**

This paper argues that using sampling-based algorithms to approximate posteriors of Bayesian neural networks (BNN) is not only beneficial but also much more realistic than is commonly thought.

As an advocate of sampling-based Bayesian methods outside of BNNs, I very much hope that the authors are correct. I personally remain sceptical, but perhaps this makes me just the right kind of person to read such a position paper (and I don't evaluate the paper based on my own standpoint of course).

**Position:**

Yes

**Position In Title:**

Yes

**Related Work:**

4

**Strengths And Weaknesses:**

The paper provides a clear, well argumented and evidence position. I didn't find any major weaknesses.

If anything perhaps one can say that the defacto benefits of sampling-based methods could be detailed a bit more. The paper focusses a lot of feasibility, which I think makes sense. Yet the "why we need this in the first place" could have taken perhaps a bit more space in the paper.

**Support:**

4

---

> ### Author Rebuttal · Authors · 2026-03-30
>
> Dear Reviewer Wk62,
>
> Thank you for your very encouraging review. We are glad you found our position clear, well-argued, and evidence-based.
>
> **Benefits of SAI.**
>
> > The defacto benefits of sampling-based methods could be detailed a bit more. The paper focusses a lot on feasibility.
>
> We agree and will happily extend the discussion on the "why" of SAI beyond feasibility. In particular, we will elaborate on the unique benefits that SAI provides: faithful uncertainty quantification that goes beyond calibrated point predictions, the ability to serve as a foundation for diverse downstream tasks (e.g., model selection, hypothesis testing, active learning), and the valuable insights SAI provides into the structure of BNN posteriors that are for the most part inaccessible through purely optimization-based methods.
>
> **Figure 3 Clarification.**
>
> > How is parallelism employed that it changes the number of forward passes?
>
> Your interpretation is exactly right. The parallel bars simply divide the number of forward passes by 10 to represent the effective throughput when using 10 cores, i.e., the number of forward passes per unit of wall-clock time. We will make this clearer in a revised figure caption. Regarding the number of posterior samples: the sample counts shown in the plot are taken directly from the respective publications. The choice of how many samples to use varies across papers. Some authors employ an early-stopping approach that monitors function-space metrics on a validation set (such as cumulative LPPD), while others make subjective choices without explicit justification. This heterogeneity is itself an argument for the standardized practices we advocate in Section 5.
>
> **Minor Points.** Thank you for noting the DE abbreviation issue and the E2E notation. We will introduce DE more prominently in the main text and spell out end-to-end in the final version (to be honest, the abbreviation was used to adhere to the page limit).
>
> ___
>
> We are grateful for your positive and constructive feedback. Please let us know if there is anything else we can clarify. Thank you again!

---

> > ### Author Rebuttal · Reviewer_Wk62 · 2026-04-01
> >
> > Thank your for your clarifying responses. I maintain my score.

---

### Official Review · Reviewer_UFxr · 2026-03-10

**Significance:** 3
**Argument Clarity:** 4
**Rating:** 6
**Confidence:** 4

**Questions:**

1. How would the authors address the concerns of the MCMC community in regards to convergence criteria and exactness in order for them to adopt the SGMCMC group of algorithms as a practical alternative?
2. The reposes section is very well done and outlines all of the major issues commonly brought up with SAI strategies. I understand that space is limited, but would the authors be able to expand upon the alternative views in the appendix along with the supporting literature for these views? Increased supporting literature for their responses would also be greatly appreciated.
3. The section on efficient storage, inference and Bayesian model averaging is insightful as there is also a wider concern in the ML community about computing resources which I believe this indirectly addresses. A potential method to aid in this would be Bayesian corsets of samples, have the authors thought about including some of this work as a suggestion in this section?

**Alternative Views Section:**

Yes

**Compliance With Llm Reviewing Policy A Conservative:**

Affirmed.

**Discussion Potential:**

4

**Final Justification:**

The authors have addressed all my concerns in their rebuttal. As a consequence, I have raised my scores for related work and my final rating for the paper.

**Paper Summary:**

The authors argue that sampling based inference (SAI) for Bayesian deep learning (BDL) is the future. They further argue against current criticism and perceived misconceptions of SAI such as time complexity, scalability and prior choices.

They also give suggestions on where future research in the BDL community should concentrate its efforts. For example, prioritizing flexible exploration of posteriors using SAI as opposed to posterior approximations, parallelized computation to reduce autocorrelations between samples, and more large scale studies of Stochastic Gradient Markov Chain Monte Carlo (SGMCMC) to share best practices, as the performance of this particular method is very hyperparameter sensitive.

The paper then moves onto suggestions for efficient sample storage and inference. Initially, all samples should be saved as hard disk memory is cheap. Improving inference at test time is more challenging, as standardized methods and comparisons for thinning samples need to be developed or used as benchmark comparisons.

Finally, the authors discuss the use of SAI in Large Language Models (LLMs). Although SAI of the full model parameters is infeasible in this scenario currently, previous work has shown that sub sampling certain layers, also known as partial Bayesian neural networks (pBNNs), still produces meaningful uncertainty quantification (UQ) estimates. This gives a pragmatic outline of how it is possible to distill UQ knowledge into foundational models and LLMs.

**Position:**

Yes

**Position In Title:**

Yes

**Related Work:**

4

**Strengths And Weaknesses:**

### Strengths
- The paper is well written and thorough.
- It addresses nearly all of the major concerns surrounding the BDL literature and a clear roadmap of how the community can move forward in this.
- It discusses key aspects that need improving in the Bayesian inference community that need to be addressed in order for SAI to be seen as a practical method in real world scenarios. E.g. using sub sampling schemes for LLMs and foundational models.
- It addresses the common alternative views and the responses are well thought out.

### Weaknesses
- Some in the MCMC may argue that the MH criteria provides convergence guarantees that SGMCMC methods do not. For example, the MH criteria ensures the algorithm targets the correct posterior and also dynamics used such as HMC obey detailed balance, where as SGHMC includes a friction parameter which violates the reversibility. The work does not address how we satisfy the potential concerns from this community.
- SGMCMC implementations also commonly use tempering in order to help explore the posterior distribution, as scaling the mini-batch gradient to approximate the full-batch has been known to cause performance issues. However, this tempering fundamentally changes the geometry of the probability distribution, so the algorithm does not necessarily target the posterior of interest. The authors have briefly touched upon the "cold posterior" effect in their work, but I believe this issue merits more discussion as it is a contentious point.

**Support:**

3

---

> ### Author Rebuttal · Authors · 2026-03-30
>
> Dear Reviewer UFxr,
>
> Thank you for your positive evaluation and constructive comments. We are glad you found the paper well written and thorough, and we particularly appreciate the "excellent" rating for discussion potential and argument clarity.
>
> **Convergence Criteria and the MCMC Community.**
>
> > How would the authors address the concerns of the MCMC community in regards to convergence criteria and exactness in order for them to adopt the SGMCMC group of algorithms?
>
> This is a very relevant question that also connects to your comment about MH adjustment and detailed balance. The core issue is a bias-variance trade-off that is specific to the BNN setting: strictly eliminating discretization error (e.g., via MH correction) incurs disproportionate inefficiencies in high dimensions, leading to extremely low acceptance rates and thus massive Monte Carlo error under any realistic computational budget. In the BNN setting, initialization error and Monte Carlo error from insufficient samples typically dominate over discretization bias when a sufficiently small step size is used. This does not hold in general MCMC, which is why the concern is understandable. We will expand the existing discussion in Section 3 to make this regime-specific argument more explicit. We are also happy to extend the paper with a dedicated critical discussion on relevant convergence diagnostics in the non-identifiable and overparameterized regime, which we currently touch upon only briefly in Section 5.4.
>
> Regarding the cold posterior effect and tempering, we agree that this merits more discussion. Tempering does change the target distribution, but recent work (Izmailov et al. 2021 and also Kapoor et al. 2022) show that much of the cold posterior effect can be attributed to interactions with data augmentation rather than fundamental misspecification. We will expand this discussion.
>
> **Expanding Alternative Views.**
>
> > Would the authors be able to expand upon the alternative views in the appendix along with the supporting literature?
>
> We will happily do this. Thank you for the excellent suggestion. We are also open to incorporating specific references you may want to suggest.
>
> **Bayesian Coresets.**
>
> > A potential method to aid in this would be Bayesian coresets of samples.
>
> Thank you for the kind words about Section 5 and this constructive suggestion. While our exposition primarily focuses on compressing and distilling posterior samples for faster and improved inference, it makes a lot of sense to also discuss approaches that make the training and sampling itself more efficient. We will include the canonical reference for Bayesian coresets (Huggins et al., 2016) and extend the discussion to dataset distillation (Wang et al., 2018) as a further dimension to increase training efficiency. This is particularly relevant for sampling, as the performance gap between full-batch and mini-batch approaches is often more pronounced than in optimization, and extending the feasible full-batch regime would be very beneficial.
>
> ___
>
> We appreciate your thoughtful and constructive review. Please let us know if there is anything else we can address. Thank you again!

---

> > ### Author Rebuttal · Reviewer_UFxr · 2026-04-01
> >
> > - Response to Convergence Criteria and MCMC Community rebuttal:
> > 	* I agree with the authors that the eliminating the discretisation error incurs a trade off resulting in low acceptance rates etc which is not generally worth it in DL scenarios. If full batch evaluation of the dataset were possible (and often not in BNN cases) then the MH correction is valuable. In a practical scenario, non-exact methods (such as SGHMC and SGLD) balance this trade off very well. I am happy to hear that the authors will include relevant literature in the revised manuscript.
> > 	* The reviewers have also pointed out the cold posterior effect can be attributed to interactions in data augmentation, among author arguments that have been postulated for this effect. This is a point of the original paper which I had forgotten and is a valid argument.
> >
> > - Response to Alternative Views rebuttal:
> > 	* I think the responses on convergence criteria from the previous chapter also relate to "Common criticism: Practitioners can never be sure whether sampling reaches a stationary distribution.". [1] would be a useful bit of literature to include in this. NUTS is considered by many to be the gold standard, and elegantly proposes an algorithm which is adaptive and still obeys detailed balance. However, it is seldom used in the BNN literature as a benchmark method because of its computational cost. If one could trade the benefits of NUTS (or adaptive HMC methods more broadly) for a small uncertainty in the targeting of a stationary distribution, this would be largely beneficial to the BDL community.
> >
> > - Response to Bayesian Coresets rebuttal:
> > 	* I agree the position posited is generally more concerned with faster and improved inference. However, test time evaluation is part of the BNN pipeline which still incurs larger computational overhead compared to SGD methods, so I thought potential improvements related to this part of the process would aid in the papers discussion. I am glad to hear the reviewers will include relevant research as I believe it will aid future readers of the paper. Deep analysis of this is not needed as it is not the focus area of the paper.
> >
> > - General Comments on the rebuttal
> > 	* Upon reading the authors responses. I believe that the upcoming included literature greatly improves upon the already very well presented manuscript. With this in mind I have revised my score.
> > 	* I would also like to emphasise to the reviewers that I enjoyed reading the manuscript.
> >
> > [1] - Matthew D. Hoffman and Andrew Gelman. The No-U-Turn Sampler: Adaptively Setting Path Lengths in Hamiltonian Monte Carlo

---

### Official Review · Reviewer_RSPb · 2026-03-18

**Significance:** 3
**Argument Clarity:** 3
**Rating:** 5
**Confidence:** 2

**Questions:**

Please see above. The criticism to the paper can be summarized as follows: are the common misconceptions really clarified or some of them are rather debatable? And in the view of that, how sure the authors are of the proposed advices in particular along the lines of abandoning parametric approaches and deployment of SVI?

**Alternative Views Section:**

Yes

**Compliance With Llm Reviewing Policy A Conservative:**

Affirmed.

**Discussion Potential:**

3

**Final Justification:**

The rebuttal has addressed my concerns -- after some smoothing of the corners, I have not objections. I checked also there were no critical issues raised in other reviews. I am happy to rise my score to "accept".

**Paper Summary:**

The paper advocates for sampling based inference (SAI) in Bayesian deep learning, counterposed to parametric approximations. The paper's position is that SAI is computationally as effective as parametric methods, while being much more flexible. The paper argues that SAI is underappreciated because of a number of (outdated) misconceptions and makes proposes what research and infrastructure improvements would help its advance and broader adoption.

**Position:**

Yes

**Position In Title:**

Yes

**Related Work:**

3

**Strengths And Weaknesses:**

## Strengths

I Believe Bayesian deep learning is one of the core topics of ICML. Advocating that it is time for fully switching the research and practice to SAI, and for wider adoption of Bayesian learning solutions, the paper picks a tough fight. By saying that, I want to emphasize that the position is non-trivial and I guess it would inspire discussions. The paper is well structured, the position is well articulated, supported by the literature and experimental data, it discusses former and current limitations, open questions, and calls for an improved community collaboration around generating, sharing, postprocessing and applying posterior samples.

## Weaknesses
I am not an expert in the field, but let me nevertheless contribute my 5 cents of feedback. As a minor point, is it obvious that $p(\mathcal{D})$ is the key challenge, especially if one is interested in drawing samples from $p(\theta | \mathcal{D})$? Then it seems to me that referring to the Laplace approximation and variational methods as MAP is confusing. Izmailov et al. 2021, cited in line 209, does not appear to discuss data augmentation.

### Prior and Posterior
It seems to me that the paper too easily dismisses the concerns about a reasonable prior suggesting that isotropic Gaussian prior suffices. This in turn suggests that the prior is not important and the posterior is largely shaped by the evidence. However, the literature on the capacity of NNs, pruning, and connectivity of well performing solutions by whole paths, if I understand correctly, suggests two things in the overparameterized mode: 1) there is high-dimensional manifold where the likelihood is close to 1 and it contains good models that generalize as well as very poor models that don't; and 2) there is still a large high-dimensional submanifold of models that generalize. So 1) implies that a prior would have a huge importance and would be detrimental for the model performance. And 2) implies that an accurate integration over all well-performing models by sampling is virtually impossible. How do the authors envision addressing these difficulties?

In the meantime, the practical field, I believed, have developed well-known workarounds. Although with no theoretical justification, but conceptually, the "prior" is given by well-performing models trained on huge data (self-supervised, masked autoencoders, CLIP, etc.), so the big data is the prior, and the "Bayesian inference" is achieved by fine-tuning on the small dataset of interest, i.e. finding the model $\theta$ that is near from the prior while also fitting the target small data, and "near" is given by SGD solutions with small learning rate.

### SAI vs Variational and Cost of the Inference
When conceptually comparing to parametric methods, the paper argues that the computation cost of SAI to generate 1000 posterior samples of one chain is comparable to the time of training the model and this comparable to obtaining the variational / Laplace approximation. However, many approximate parametric methods are designed such that they can be applied in practice at the computation cost of approximately one forward pass, which makes them acceptable in practice for e.g. uncertainty quantification. Clearly, the flexibility and possibly accuracy of SAI would be a tradeoff for the cost. If it is not posed to strictly dominate in both the speed and accuracy, is it reasonable to call for abandoning the research on parametric methods and approximate predictives?

### Value of SAI/BDL in practice
It seems to me the paper calls for wide-spread adoption of SAI in practice, without openly facing the practicality questions.
The paper cites experimental results on some classical datasets and compares SAI with e.g. Deep Ensembles and Laplace in terms of the performance of the predictive distribution. However, for typical datasets the numbers are substantially lower than a single point-estimated model can achieve (with proper architecture facilitating good-conditioning for optimization, initialization, augmentation, regularization or BN).  Some of these practices have nothing to do with BDL and are rather incompatible with it. But practically we have that BDL can bring a bit of improvement with huge inference overheads vs substantially more improvement via heuristic deep learning and little to no inference overhead.
A similar question arises for the uncertainty quantification. To assess its advantage, one need to utilize it in some downstream task, e.g. statistical decision making (recognition with rejection, asymmetric decision cots). In this specific case, the risk of the decision making will depend on both the sharpness of the predictor and its reliability (uncertainty) and one may discover again that a strong-performing point estimate with a calibrated post-processing of predictive probabilities would perform better that a worse-performing model with a perfectly accurate uncertainty quantification. So is deployment really on the agenda? In which applications?

**Support:**

3

---

> ### Author Rebuttal · Authors · 2026-03-30
>
> Dear Reviewer RSPb,
>
> Thank you for your review. We are grateful for the time invested and the many constructive angles you raise, which genuinely enrich our position.
>
> **Terminology**
> > It seems to me that referring to the Laplace approximation and variational methods as MAP is confusing.
>
> Fair point. We do not mean to equate these methods with MAP estimation per se, but rather to highlight that they cast the inference problem into an optimization problem, often centered around a mode of the posterior. We will clarify this distinction in the revised manuscript.
>
> **Izmailov et al. (2021) & Data Augmentation**
> > Izmailov et al. 2021, cited in line 209, does not appear to discuss data augmentation.
>
> The reviewer might have overlooked Section 7.1 in Izmailov et al. (2021), where data augmentation is discussed extensively. We will add a more specific pointer.
>
> **Prior and Posterior**
> > The paper too easily dismisses the concerns about a reasonable prior suggesting that isotropic Gaussian prior suffices. [...]
>
> We do not argue that priors are unimportant. Rather, standard isotropic Gaussian priors work well in practice (see e.g. Paulin et al. 2025 or Duffield et al. 2025) and constitute a meaningful choice for BNNs due to their interaction with overparameterization and inductive biases (Kobialka et al., 2026) in the absence of additional knowledge.
>
> We also agree that using a pretrained model as prior with fine-tuning data as likelihood is practical in continual learning, and will incorporate this. However, our position focuses on the algorithmic challenge and common misconceptions. Like in classical NN training, where optimization and transfer learning are different worlds, we see the algorithmic challenge in Bayesian model training and knowledge transfer (e.g. via priors) as separate topics, with our position focusing on the first.
>
> > [...] an accurate integration over all well-performing models by sampling is virtually impossible. How do the authors envision addressing these difficulties?
>
> This is where the hybrid approach, which is increasingly advocated in the literature, becomes essential. By warm-starting chains from optimized solutions and performing local exploration, one captures meaningful local posterior structure rather than attempting full integration. Tab 1 and Fig 2 show that even this localized approximation consistently outperforms optimization-based alternatives. Full integration is impossible, but our results and recent literature show that sampling is often notably better suited for this problem and, contrary to common belief, neither infeasible nor much slower in practice.
>
> **SAI vs. Parametric Methods & Inference Cost**
> > However, many approximate parametric methods are designed such that they can be applied in practice at the computation cost of approximately one forward pass.
>
> Our position is not a general rejection of parametric methods, but one must distinguish training and inference cost. Many variational and approximate methods are not notably faster, if at all, **at training time** compared to SAI (Fig. 2). **At inference time**, where some approximate parametric methods only require one forward pass, SAI is definitely slower, requiring as many forward passes as samples. But this is exactly the point of our Section 5: Sampling is scalable and better at training. So we can get notably better posterior approximations for most use cases. But we need to make inference efficient, possibly by adopting strategies of parametric models, to make sampling accessible.
>
> **Practical Deployment**
> > One may discover again that a strong-performing point estimate with a calibrated post-processing of predictive probabilities would perform better [...] So is deployment really on the agenda?
>
> A fair point. First, recent SAI-based BNNs can match or exceed strong baselines (e.g, [1], comparing BNNs to TabPFNv2). So there seems to be also a merit just from a performance point of view. Second, we argue that most classic DL benchmarks are not well-suited for evaluating UQ ([2]). Third, calibration in predictive space and UQ in parameter space are conceptually distinct and should not be conflated. The latter captures the model’s uncertainty due to lack of information, whereas the first “just” matches the predictive distribution with the empirically observed one. Lastly, effective hybrid sampling strategies build on strong optimization-based solutions and add flexible local exploration, complementing rather than conflicting with standard DL practices. Deployment is reasonable wherever the latency overhead can be addressed.
>
> - [1] Arvanitis et al., bde: A Python Package for Bayesian Deep Ensembles, 2026
> - [2] Kapoor et al., On Uncertainty, Tempering, and Data Augmentation in Bayesian Classification, 2022
> ___
> We believe the reviewer's suggestions have provided valuable additions that will strengthen the paper. Please let us know if there are any further points we can clarify. Thanks again!

---

> > ### Author Rebuttal · Reviewer_RSPb · 2026-04-03
> >
> > I thank the authors for the detailed response. At this point I do not have further questions.

---

### Decision · Program_Chairs · 2026-04-30

**Decision:**

Accept (spotlight)

**Comment:**

This is an outstanding paper that makes a surprising and very important point.  It supports the point with sound logical arguments and technical details.  There was outstanding discussion between the authors and the reviewers, and the reviewers became even more positive from the discussion.